# Age-related differences in trust beliefs during middle childhood: Downward-extension and validation of the general trust scale

**Alex R. Wheeler**[1]*, **Donna M. Bayliss**[1‡], **Jeneva Ohan**[1,2‡]

1 School of Psychological Science, University of Western Australia, Perth, Western Australia, Australia,
2 Youth Mental Health, Telethon Kids Institute, Perth, Western Australia, Australia

‡ These authors are joint senior authors on this work.
* alex.wheeler@research.uwa.edu.au

## Abstract

There are conflicting suggestions concerning the developmental trend of trust beliefs during middle childhood. Across three studies, the current research developed a brief measure of child general trust beliefs, as well as child measures of trust in peers and online, and examined age-related differences in these beliefs. Study 1 explored the appropriateness of downward extending the General Trust Scale. Studies 2 and 3 developed the child version of this scale and adapted the target of trust to construct two additional scales measuring trust beliefs in peers and online. These studies also provide evidence of the psychometric quality of the scales, and that trust beliefs are positively associated with friendship quality and psychosocial well-being outcomes in children. In addition, Study 3 demonstrated small age-related decreases in general and peer trust. This finding suggests children may become more discerning during middle childhood. Implications of these age-related differences and the use of these novel scales is discussed.

## Introduction

Believing in the trustworthiness of other people is important to children's development, with lower levels of trust associated with loneliness [1–3], aggression [4,5], long-term bullying, and problematic friendships [6]. There is also evidence that high levels of trust can be as disadvantageous as low trust; both are associated with lower self-perceived [7] and observed [8] social acceptance and increased social distress. Trust in others is often described as general trust, the belief most other people will treat each other (including the trustor) benevolently [9,10] In children, trust in others is also conceptualised as peer trust – the belief that friends or classmates are trustworthy [5,7,11,12]. However, whether these two forms of trust beliefs are distinct and differentially predict psychosocial indicators is uncertain, impeding understanding of the factors that support well-being during childhood development. Additionally, it

**Data availability statement:** The materials and analysis code for all the studies in this report cannot be shared publicly due to ethical reasons related to the confidential storage of sensitive information about a vulnerable population, and the wording of the informed consent document requiring only the research team access any data. Data requests should be directed to the Human Research Ethics Office of the University of Western Australia (human-ethics@uwa.edu.au).

**Funding:** The author(s) received no specific funding for this work.

**Competing interests:** The authors have declared that no competing interests exist.

is unclear whether children's trust in other people increases or decreases through middle childhood, when peer relationships become increasingly complex, supportive, and intimate [13]. Further, evidence suggests that children begin to interact with others not just in person, but also online, younger than ever before [14], yet no child trust measures are available that investigate possible differences in trust in others in online contexts.

One of the most consistently demonstrated relationships between trust beliefs and psychosocial well-being in middle childhood is that with loneliness, which is negatively correlated with both general trust [3] and peer trust [1,15]. Trust in peers also has a quadratic association with peer victimisation and quality of social interactions, such that children with both very high and very low levels of trust beliefs in peers experience increased peer rejection, decreased peer acceptance, and higher levels of distress and indirect aggression during social interactions [8], as well as poorer quality of friendships and social acceptance [7]. These social outcomes clearly relate to peer trust as they are determined by the nature of interactions with peers. However, whether general trust has a similar relationship with these or other friendship and psychosocial well-being outcomes is yet to be established.

Some insight into the unique relationships that general and peer trust might exhibit with psychosocial indicators comes from research into associations with prosocial behaviour. While the findings are limited, general trust is associated with prosocial behaviours, such as helpfulness towards classmates, in middle childhood [16]. Associations have also been demonstrated in late childhood and adolescence [17], with this association described as one that develops as children age [18,19]. Conversely, peer trust appears to exhibit no such relationship with prosocial behaviours, demonstrated by the lack of significant association between measures of trust in classmates and teacher-rated prosocial behaviour [20], or peer trust with the frequency of prosocial behaviours observed at school [8]. Together, these findings suggest prosociality may be an important association unique to general trust in middle childhood.

Despite the importance of general trust for child psychosocial development, there is uncertainty as to whether and how it changes with age. On one hand, rapid age-related increases in selectivity of whom to trust and sensitivity to situational cues from early through middle childhood may suggest a broader trend indicating decreased general trust. On the other, research suggests trust behaviours increase with age from middle childhood to early adulthood. Evidence suggesting an age-related decrease in general trust derives from research into trust in testimony (concerning the veracity of claims) and selective trust (deciding whom to trust). While young children have a general bias to trust new information from adults [21], children can learn to evaluate informants based on their personality or history of accuracy [22]. For example, in one study, 4-year-old children were far less likely than 3-year-olds to trust information presented by an adult who was repeatedly deceptive [21]. By age 7, the degree or frequency of inaccuracy of an informant is considered, as children learn to doubt more appropriately in response to small and large errors [23]. Older children are also less trusting of testimony presented by non-experts. In a study comparing 3- and 8-year-old's trust in information presented as expertise [24], 8-year-old children

were more likely to trust information when presented by a domain-relevant expert, and more likely than the younger group to distrust testimony presented by one whose expertise was in an unrelated domain (e.g., zookeepers presenting their testimony on car speeds).

Selective trust research also demonstrates that young children prefer information from a trustworthy adult, suggesting they develop consideration for the reliability of an informant's previous claims [25]. This body of work also suggests that despite the general bias to trust testimony and continue to believe information that has demonstrated inaccuracy [21], children develop a heuristic from age three to trust the more reliable informant of those whose testimonies diverge [26]. Finally, older children are less trustful of distorted claims by age 9, thought to be due to better critical thinking skills that improve with age [27]. Taken together, these studies demonstrating improved abilities to detect untrustworthy sources or information in specific contexts, and while they do not measure changes in general trust specifically, they may suggest a broader trend of decreases in general trust throughout childhood.

Next, we review evidence suggesting the inverse trend. For example, Sutter and Kocher [28] investigated the development of trust behaviours in a large-scale cross-sectional study using the Investment Game [29], a procedure that estimates trust levels based on the amount of money a player invests in their partner. Sutter and Kocher [28] found that the proportion of funds invested increased with age from 8 years to early adulthood, suggesting an increase in trust, then plateaued through to retirement age. However, some researchers highlight the importance of distinguishing trust as an economic decision-making behaviour such as this from trust beliefs that may become more selective with age, as described above [30]. Also, the aspect of cooperation involved in the Investment Game may rely on prosocial behaviours independent of trust in the partner player [31]. While this behavioural measure of trust may not be directly comparable with measures of trust beliefs, Poulin and Haase [32] demonstrated that trust beliefs also increase from age 14 years into adulthood using longitudinal research. Nonetheless, the question of whether levels of trust in others differ with age before this adolescent stage remains unresolved.

## The construct of general trust

While the research on age-related differences in trust concerns trust in others broadly, the concept of general trust itself is important to define. The conceptualisation of general trust as a belief – in contrast with other conceptualisations akin to a personality trait, e.g., propensity to trust [33] – is supported by research demonstrating this belief is fragile. That is, general trust is quickly replaced by a specific judgment when relevant information is available suggesting trustworthiness or untrustworthiness is appropriate, particularly among those high in general trust [34]. Fundamentally then, general trust is an initial expectation of whether others will be trustworthy in the absence of cues that inform a decision of whether to trust a specific entity. A comprehensive review proposed that in the absence of such cues, trustworthiness expectations are governed by prior trust experiences and projecting one's own trustworthiness onto others [35]. There has been little consensus on whether this belief is unidimensional [36,37] or multidimensional [16,38]. However, measuring this belief using a broad general target (e.g., 'most people') or through initial trust in strangers is considered fundamental to accurate conceptualisation of general trust [35,39]. Here, we define general trust as the belief, attitude, or expectation concerning the likelihood that the actions of other individuals will serve the actor's interests in situations wherein the outcome is uncertain [9,34].

In child research, a different framework to understand trust in others has been the primary focus of investigation. Rotenberg et al [16] described a Bases Domains Targets (BDT) framework for understanding trust in others, which suggests trust beliefs in others are comprised of three fundamental bases: reliability, emotional trust, and honesty. This framework has been validated in both middle and early childhood [40], but has not been investigated in older children or in adult trust research. This lack of research across the lifespan makes it difficult to determine the true pattern of age-related differences in trust. Further, this framework differs somewhat from the conceptualisation of general trust investigated more broadly in adult populations. In contrast to the measures of general trust in adults that concern general targets [10] or strangers [39],

the BDT framework suggests seemingly specific targets such as mothers and fathers to represent part of a collection of more generalised targets [16]. Measures based on this framework estimate generalised trust in others as a sum of trust across these targets that vary in specificity and familiarity. To better understand the function of general trust across the lifespan, a measure of childhood trust in a more general target is required.

## Online contexts

Child trust research needs more than ever to address the context of trust in a digital world. Children spend more time online each year. Whereas young children tend to use digital technology individually, or are not otherwise using social media [14], children often interact socially on digital devices by age 8 years, particularly through gaming platforms [41]. During middle childhood, as many as 17% of British children are estimated to have played games online with people they have never met [42]. The pervasiveness of online social interactions for today's (and future) children is clear; for example, more Australian households with children under the age of 15 years have internet access (97%) than do those without children (82%), 14% of children had been exposed to inappropriate material online, and more than 90% of children aged 9–11 years accessed the internet at home [43]. Since these data were collected, digital interaction has arguably become more ubiquitous for children, particularly due to remote learning in response to COVID-19 outbreaks, when children may have been accessing classes on digital devices unsupervised [44]. Without in-person interaction where children can experience trustworthiness cues through nonverbal communication and others' interpretation or guidance, it may be difficult to understand another's intentions. As a result, researchers are increasingly interested in how trust might differ in an online context [45].

While research has begun to investigate the nature of trusting others online in adults [46], there is a paucity of research investigating this context in children. One important trait that may shed light on online trust interactions is risk-aversion. Risk-aversion is thought to be more salient when trusting strangers than familiar targets such as family members [35]. In online interactions, risk-aversion may be heightened due to uncertainty and unpredictability about the target of trust, resulting in lower trust during online interactions. However, children are usually less risk-averse and less rational in their decisions involving risk than adults [47], so would likely not show similar reductions in trust online, leaving them more vulnerable in this context. Communication online has become inherent to daily life, so understanding the nature of children's trust online is of great importance. However, no child trust measures are currently available that assess online contexts.

## The present research

Research has yet to determine whether children's levels of general trust increase through middle childhood in the way suggested by behavioural research [28] and in adolescents [32], or decrease as does trust in testimony during early childhood [21,24]. Further, clarity is needed as to whether general and peer trust are distinct and show unique relationships with psychosocial indicators. There is also a need to measure children's trust in online contexts and understand how it functions relative to general and peer trust.

To address these questions, we downward-extended the Yamagishi and Yamagishi [10] General Trust Scale (GTS) for use in children as young as 8 years old and adapted the modified scale to create versions assessing peer and online trust. Across three studies, we investigated the psychometric properties of this downward-extended GTS, presented evidence for the validity of the alternate versions, and examined age-related differences in levels of general trust beliefs, peer trust beliefs and online trust during middle childhood, and the relationship between trust beliefs and psychosocial indicators. In the first study, we examined the appropriateness of the language in the GTS for use with children to ensure its suitability for downward extension. The second study investigated the convergent validity of the downward-extended GTS (modified to increase readability and clarity for child readers) and its alternate versions by examining relationships with an existing measure of child trust beliefs and measures of loneliness and prosocial behaviour. In the third study, we used the modified GTS to examine age-related differences in the general trust beliefs of children aged 8–12 years, and the relationship

between general trust and children's well-being, friendship, and other social outcomes, including whether any of these relationships are curvilinear. This final study also examined the internal structure through a confirmatory factor analysis (CFA), and further explored the evidence for convergent validity of the modified GTS and its alternate scales.

## Study 1A

Children learn to understand most abstract words (such as 'trust') later than concrete words [48]. In adults, many popular and recent trust measures administer items that gauge respondents' trust in others directly by using the word 'trust' in the item/s [10,36,37,49]. While one recently constructed child measure does ask child respondents how much they trust others directly [17], the most widely used measures of trust beliefs in children instead require respondents to judge the likelihood that others will behave in accordance with their word or the respondents' best interests [16,40]. This wording may help ensure clarity among child respondents through use of more concrete language than the direct measures. However, whether children benefit from this concrete language in the context of trust measures has not been established. In fact, children have demonstrated understanding of various abstract words with positive valence – including the word 'trust' [50] – by age 8–9 years, and understanding of other abstract words (including both positively and neutrally valenced) by age 10–11 years [51]. If children do have sufficient understanding of the abstract construct of 'trust', it would allow the administration of direct trust measures. The similarity of this measurement approach to adult instruments would enable empirical comparison between adult and child populations.

To confirm that direct measures of general trust are appropriate to administer to children, this study investigated whether children understood the meaning of the word 'trust'. By asking children to define this word without additional prompts, we aimed to assess whether children of this age have sufficient understanding of the construct to be administered items incorporating the word 'trust'. Responses were scored against the definition of general trust provided above, and whether they referred to any of the three bases of trust beliefs described by Rotenberg et al. [16]. By doing this, Study 1A aimed to confirm the suitability of one of the most widely administered adult trust measures, the GTS [10], for downward-extension for children aged 8 years and older. It was hypothesised that as a positively valenced abstract word, most of the children would demonstrate an understanding of the word 'trust' such that they could provide a sufficient definition. Further, a logistic regression was conducted to determine whether or not this ability exhibits any effect of age in middle childhood.

## Methods

### Participants

Ninety-one Australian children ($n = 43$ (46.67%) female; $n = 48$ male) were recruited, of which parents identified 68 (74.7%) as Caucasian or White, 13 as Asian, six as mixed, two as Aboriginal, and two who declined to answer. The mean age was 10 years and 3 months (range = 8;0–12;10, $SD = 1.39$), with 97.8% indicating they lived in a household that had been able to meet all essential household expenses within the last year (the other two parents of these participants declined to respond to this demographic question). Participants reported their financial circumstances as comfortable ($n = 41$), tight ($n = 47$), and struggling ($n = 2$). All studies reported in this report were approved by the University of Western Australia Human Research Ethics Office (2020/ET000298). Adult participants provided informed written consent and children provided informed written assent before participating. Participants were recruited between 31 May 2021 and 22 August 2021 and were offered AUD$15 as reimbursement for their time and travel costs.

### Measures

The children participated in person at the University or via a Zoom video-call with an experimenter. Children were verbally asked a prompt question 'What does trust mean?'. The prompt was used to elicit open-ended responses from children about their understanding of the concept of trust without further clarification by the experimenter.

## Procedure

An experimenter asked the prompt question verbally upon participants completing the questionnaire materials administered in Study 2 and recorded the responses verbatim. The measure from this study was presented after the measures from Study 2 to prevent possible priming on the nature of trust that may have influenced participant responses to the scale development items. If children stopped speaking abruptly, trailed off before completing their thought, or started to provide a partially accurate definition before indicating they were finished responding, the experimenter asked up to two of the following neutral prompt questions as appropriate: 'Can you tell me anything more about it?', 'Is there anything else you can tell me about what trust means in your opinion?', or, when children started their response with an example, 'Is there anything different to that example you want to tell me about what trust means?'.

## Analytical procedure

A single author coded the definition responses dichotomously to indicate whether responses sufficiently described general trust, according to the definition provided above. For example, responses describing trust as believing people will be dependable, or will help you by doing what they say they will, were coded as aligning with the definition (a codebook outlining this process is described in S1 Appendix). To examine whether potential developmental or educational changes resulted in a greater chance of older children defining trust sufficiently [51], a logistic regression was conducted on this dichotomous variable as the proportion of definitions coded as sufficient, predicted by age in months. Further, to explore the aspects of this definition and of the theoretical BDT framework [16] that participants mentioned in their responses, additional dichotomous variables were scored for each of the following categories: whether participants described trust in terms of a belief concerning beneficial treatment, whether they acknowledged risk or uncertainty, and whether they described it in terms of each of the three BDT framework bases of trust. Analyses for all studies were conducted in SPSS 27.0.1 except where otherwise indicated.

## Results

Descriptive statistics detailing the proportion of participants who provided a sufficient definition are presented in Table 1. As hypothesised, most participants defined the word 'trust' sufficiently, overall and within each age group. A logistic regression revealed that age was not a significant predictor of whether participants met the definition criteria ($\chi^2(1) = 2.87$, $p = .09$, OR = 1.03, 95% CI [0.99, 1.07]). Most children mentioned that a trustee acts in some way that benefits the trustor, followed by reliability, emotional trust, acknowledging the context of uncertainty, and with the fewest children mentioning an aspect of honesty.

**Table 1. Proportion of child participants who defined trust, by age and component described.**

| Age Group | Sufficient Definition | Beneficial Treatment[a] | Uncertainty/Risk[b] | Reliability | Honesty | Emotional Trust |
|---|---|---|---|---|---|---|
| 8 years old | 0.77 | 0.73 | 0.14 | 0.36 | 0.14 | 0.45 |
| 9 years old | 0.79 | 0.42 | 0.05 | 0.42 | 0.11 | 0.32 |
| 10 years old | 0.89 | 0.83 | 0.28 | 0.83 | 0.11 | 0.39 |
| 11 years old | 0.83 | 0.67 | 0.11 | 0.44 | 0.11 | 0.67 |
| 12 years old | 0.93 | 0.57 | 0.21 | 0.71 | 0.14 | 0.43 |
| Overall | 0.84 | 0.64 | 0.16 | 0.56 | 0.12 | 0.45 |

[a]Definition described 'trust' as an attitude and/or belief that the target of trust will act in some way beneficial to the trustor

[b]Definition acknowledged 'trust' is made in the context of uncertainty/risk

## Discussion

As hypothesised, most children demonstrated an understanding of the concept of trust that was consistent with the academic conceptualisation of the term. This result is consistent with Ponari et al.'s [51] findings that 'trust', among other positively valenced words, is understood by children as young as eight years, despite being an abstract concept. The non-significant regression suggests age is not likely a predictor of children's understanding in middle childhood, perhaps because trust is already understood sufficiently by age 8 years. Furthermore, the high proportions overall and among the younger age groups suggest that children as young as eight years are highly likely to be able to understand the construct well enough to answer direct questions concerning their level of trust in others. It is also worth noting that children generally develop receptive language skills faster than expressive [52–54]. Taken together, these results strongly suggest that 8-year-old children understand what trust means sufficiently to apply it to their interpersonal context.

This approach to assessing children's conceptual understanding may be limited by scoring it against academic, rather than vernacular definitions of the term. Vernacular understandings of concepts may not always align with researchers' conceptualisations, though it seems likely there is some age by which a common understanding of the word 'trust' is ubiquitous. A direct comparison of child definitions of trust with those of adult respondents would have greater ecological validity and may yield better insight into whether children demonstrate a similar degree of conceptual understanding to adult respondents when scoring their responses against an academic definition. Unfortunately, no such study exists. That is, while measures of general trust in adults have been administered asking directly about levels of trust for several decades [10,36], there has been no investigation into whether respondents conceptualise the word similarly to researchers.

Another limitation of the approach in Study 1A was that children answered the prompt question upon completion of the measures used in Study 2. While the presentation order was chosen to minimise influence on responses to scale development items in Study 2, these measures included questionnaires assessing levels of trust and may have primed the children to consider which contexts involve trusting others in Study 1A. For instance, the Rotenberg, Fox, et al. [16] measure describes a series of scenarios in which children are asked to judge the trustworthiness of parents, friends, and teachers. To address these limitations, Study 1B recruited an adult sample to compare their response pattern to the child participants in Study 1A. This adult study additionally manipulated whether the prompt to define trust was presented before or after participants completed the general trust measures, to determine whether they would be primed to better define the term 'trust'.

## Study 1B

To determine whether the definitions children provided in Study 1A were congruent with the understanding of trust among the adult population, an adult sample was recruited who completed a similar procedure to the child participants. Presentation order was manipulated such that the adults were asked to define trust before ('pre-survey' condition) or after ('post-survey' condition) completing the same trust questionnaires the children were administered. To examine whether the completion of trust scales primed participants to define the term, a Pearson's chi-square test was conducted to examine the difference in the proportions of participants who defined trust sufficiently between conditions. A similar analysis was also conducted to investigate possible differences between the adult and child samples overall.

## Methods

### Participants

Australian undergraduate psychology students ($N = 52$; $n = 40$ (76.9%) female; $n = 12$ male) participated in the research for course credit, of which 35 (67.3%) identified as Caucasian, 12 as Asian, one as both, one as both Aboriginal and Caucasian, one as African, one as Middle Eastern, and one as Indian. Participant ages ranged from 18 to 60 ($M = 21.4$; $SD = 6.88$), with 90.4% indicating they lived in a household that had been able to meet all essential household expenses

within the last year. Participants reported their financial circumstances as comfortable ($n=23$), tight ($n=25$), and struggling ($n=4$). Participants provided informed written consent before participating. Participants were recruited between 11 August 2022 and 21 October 2022.

## Measures

To determine whether the presentation order of trust questionnaires led to a priming effect on the ability to define trust, participants completed the same materials as the child participants in Study 1A apart from the measures of loneliness or prosocial behaviour. The prompt question and available follow-up questions were identical in wording and administration to that presented in Study 1A.

## Procedure

Participants were randomly assigned into the pre-survey and post-survey conditions. In the pre-survey condition, participants were first asked to define trust, and then completed the trust questionnaires outlined in Study 2 in the same order as the child participants. The post-survey condition followed the same presentation order as in the child sample, with the prompt question administered following the completion of all trust questionnaires.

## Analytical procedure

The proportion of responses that aligned with the definition of trust were coded using the same process as in Study 1A. Differences between the pre-and post-survey conditions were investigated using a 2 x 2 Pearson's chi-square test to determine any effect of condition on the ability to define trust. The two conditions were then collapsed into a single adult sample for comparison with the child sample from Study 1A using a Pearson's chi-square test.

## Results

The proportion of adult participants who provided sufficient definitions of 'trust' is presented in Table 2. Most participants defined the word 'trust' in line with this definition. Comparing the pre- and post-survey conditions, a chi-square test revealed no effect of condition ($\chi^2(1, N=52) =.22, p=.64, \varphi=-.07$), such that providing a definition of 'trust' before or after completing questionnaires yielded no difference in the ability to define the construct. Three participants in the post-survey and two in the pre-survey condition did not provide sufficient definitions of the construct.

Further, a 2 x 2 Pearson's chi-square analysis comparing the definitions provided by the full adult and child samples revealed no statistically significant effect of group ($\chi^2(1, N=143) = 1.30, p=.25, \varphi=-.10$), suggesting no difference in the proportion of the child participants (from Study 1A) versus adult participants who sufficiently defined the term 'trust'.

Table 2 shows adult participants were most likely to describe trust with respect to being treated beneficially, and least likely to mention honesty. These ranks were identical to the child sample in Study 1A.

**Table 2. Proportion of adult participants who defined trust, by condition and component described.**

| Condition | Sufficient Definition | Beneficial Treatment[a] | Uncertainty/ Risk[b] | Reliability | Honesty | Emotional Trust |
|---|---|---|---|---|---|---|
| Pre-Survey | 0.92 | 0.73 | 0.27 | 0.58 | 0.12 | 0.58 |
| Post-Survey | 0.88 | 0.69 | 0.27 | 0.62 | 0.27 | 0.42 |
| Overall | 0.90 | 0.71 | 0.27 | 0.60 | 0.19 | 0.50 |

[a]Definition described 'trust' as an attitude and/or belief that the target of trust will act in some way beneficial to the trustor

[b]Definition acknowledged 'trust' is made in the context of uncertainty/risk

## Discussion

Similarly to the child sample in Study 1A, most adult participants demonstrated an understanding of the concept of trust, with a similar proportion to those child participants. Completing measures of general trust did not prime participants to better define the term 'trust'. This finding suggests that the results of Study 1A may not have been limited by presenting the prompt question after the other questionnaire measures. Further, the similarity between the adult and child participants suggests that children as young as 8 years are able to define the word 'trust' with no less conceptual accuracy than adults. While undergraduates typically have better linguistic understanding than the general population [55], limiting the generalisability of this finding, the similarity between educated adults and children further suggests evidence for the validity of using a downward-extension of the adult GTS measure.

## Study 2

To assess how trust beliefs change with age during middle childhood, and how these beliefs relate to other important psychosocial indicators, an age-appropriate measure of general trust is needed. We first investigated the available child general trust measures, identifying two available for children of reading age. The Children's Generalized Trust Beliefs Scale (CGTBS) is lengthy at 24 items and has reported low subscale internal consistency coefficients [16]. The trust subscale of the Holistic Skills Assessment (HSA) is reliable ($\omega = .77$), and demonstrates convergent validity with prosocial behaviour [17]. However, two of only three items in the scale ("I trust other people" and "Most people can be trusted") appear arguably indistinguishable for children to discriminate, and the third has questionable face validity. Further, neither scale was the ideal candidate to modify to measure children's trust in an online target.

As such, we chose to downward-extend an established adult measure of general trust. Considering a preference for brief, reliable measures, we opted to modify the five-item GTS [10]. While a recently updated version of the scale is available – the Inclusive General Trust Scale (IGTS) [34]– the newer subscale is composed of lengthier items than the original GTS, which appeared too complex to reword into age-appropriate language while still retaining their conceptual meaning and face validity. The original GTS was retained as a subscale in the IGTS, and the IGTS has notably weaker internal consistency than the GTS [10,34], so we opted to downward-extend the original measure. Importantly, while the IGTS predicts economic trust behaviours better than the GTS [34], both scales demonstrate significant correlations with these behaviours, and the GTS continues to be used as a briefer measure in published research [56].

To create the downward-extended version of the GTS [10] – the General Trust Scale for Children (GTS-C) – we simplified the language of the GTS to ensure readability for respondents as young as 8 years. We then further modified the wording to create alternate scales assessing targets of trust online (General Online Trust Scale for Children; GOTS-C) and in peers (General Peer Trust Scale for Children; GPTS-C). This study aimed to investigate the psychometric properties of the resultant scales. Namely, we hypothesised that the GTS-C and GPTS-C would demonstrate positive correlations with an existing child measure of trust beliefs in others, with prosocial behaviour [16], and with loneliness [3].

## Methods

### Participants

The participants were the same sample as in Study 1A, who completed these materials prior to answering the prompt question described in Study 1A. A power analysis conducted in G*Power 3.1 [57] determined a minimum sample size of 44 would be required to detect an effect size demonstrated between the CGTBS and a prosocial behaviour in prior research [16] with 80% power and a two-sided significance level of.05. A convenience sample of at least double this requirement was recruited in an attempt to reduce familywise error.

## Measures

We simplified the wording of the five-item version of the GTS [10] to ensure Flesch-Kincaid readability for a Year 3 or eight-year-old level. For example, the item, "Most people are basically honest" was modified to, "I think most people tell the truth". This simplification also resulted in a five-point response scale (from "Disagree a lot" to "Agree a lot"). Two subject matter experts reviewed the modified items to ensure simplified items retained the original meaning of the adult scale and were in line with our definition of general trust. Additionally, we consulted schoolteachers to determine whether the concepts and language would be suitable to administer to the target population, and made minor edits based on this consultation process. This process resulted in the General Trust Scale for Children (GTS-C).

To create a trust in peers scale, the five items of the GTS-C were modified to replace the target of trust with "children in my year" (used to refer to children in the same grade/year at school), to assess children's trust in others their own age. The resulting scale was named the General Peer Trust Scale for Children (GPTS-C). Due to one first-person item in the GTS [10], the GPTS-C contained the same final item ("I am a trusting person") as the GTS-C.

Additionally, a five-item online trust in others subscale was created by replacing the target of trust with "people online" to create the General Online Trust Scale for Children (GOTS-C). Due to one first-person item in the GTS [10], the GOTS-C contained the same final item ("I am a trusting person") as the GTS-C.

## Measures for validity analyses

To assess convergent validity with a similar existing measure, we administered the Children's Generalized Trust Beliefs Scale ($\alpha = .76$), a measure of children's trust in others that provides a composite score of general trust by averaging trust across several short situations describing familiar targets (mother, father, teacher, peer) [16]. The 24-item scale has three subscales assessing reliability, honesty and emotional trust ($\alpha = .67, .65, 62$, respectively), scored on a five-point likelihood scale. A sample item is "Sarah's Mother said that if she cleans her room she can go to bed half-an-hour later. Sarah cleans her room. How likely is it that Sarah's Mother will let Sarah go to bed half-an-hour later?".

The six-item Prosocial Behaviour subscale of the Social-Emotional and Character Development Scale (PBS; $\alpha = .74$—.86) was administered to assess convergent validity between children's general trust and helpful behaviours toward others [58]. Items are rated with a simple response format ("NO!", "no", "yes", "YES!"), designed to be suitable for younger children to self-report.

Convergent validity with loneliness was assessed with the Loneliness and Social Dissatisfaction Scale (LSDS) [59]. We administered a four-item version with demonstrated internal consistency ($\alpha = .87$) [7], namely "I have no one to talk to", "I feel alone", "I feel left out of things", and "I am lonely" (rated on a five-point scale from "Not true at all" to "Always true").

## Procedure

The questionnaires were administered to children via Qualtrics using standard computers. Some children participated in-person, with a research facilitator sitting beside the child. Alternatively, children participated via Zoom in a digital meeting, for which a facilitator provided a link to the Qualtrics questionnaires. Before completing the online trust scale, children were asked whether they had ever communicated with strangers online, and only those who indicated they had ($n = 37$) were invited to complete the online subscale. The total session lasted approximately 30 minutes.

## Analytical procedure

Sum scores were generated for each of the measures of trust in others and psychosocial indicators. Data were screened and checked for missing values before examining for influential cases using the interquartile range rule method [60]. Normality was examined by calculating the ratio of skewness and kurtosis to each of its respective standard errors. Convergent validity was assessed by examining one-tailed correlations between each of the modified versions of the GTS with

the CGTBS, each other, and the measures of psychosocial indicators. Internal consistency of the GTS-C and its modified versions were examined using McDonald's omegas. Values above.70 were considered acceptable indicators of internal consistency [61]. Correlation effect sizes were deemed large if >.30, medium if >.20, and small if >.10 [62]. The analyses were conducted using SPSS 27.0.1, except for calculating internal consistency coefficients using jamovi [63].

## Results

### Reliability

Data were initially screened to ensure that parent responses were available for all completed child data. Responses were included for analysis when linked data were available for both the parent and child participants. Data were screened prior to analysis and contained no missing values. The ratios of skewness and kurtosis to their respective standard error scores were calculated for each variable of interest. Spearman's correlations were conducted on LSDS and PBS scores, which violated the assumption of normality, as demonstrated by ratios larger than |1.96| – the cut-off recommended for samples smaller than 200 [64].

The GTS-C demonstrated low internal consistency ($\omega = .68$), but all item-total correlations exceeded.40, so were retained. The GPTS-C ($\omega = .77$) and GOTS-C ($\omega = .83$) both demonstrated acceptable internal consistency. We also examined the reliability of four-item versions of the GPTS-C and GOTS-C by removing the fifth item of each that was repeated from the GTS-C (i.e., "I am a trusting person"), as it did not indicate an external target of trust. These four-item scales maintained acceptable internal consistency ($\omega = .75, .80$, for the GPTS-C and GOTS-C, respectively). For all versions of each scale, item-total correlations exceeded.40.

While the five-item GPTS-C and GOTS-C demonstrated nominally higher internal consistency than the four-item scales, the fifth item was not modified from the GTS-C that was designed to measure a different target to either scale. As such, we determined that measures of peer and online trust would not benefit from this item and it was dropped from each scale, resulting in four-item versions of the GPTS-C and GOTS-C (items for the final versions of all scales are available in Tables 3, 4, and 5, respectively). Descriptive statistics for all measures are presented in Table 6.

### Convergent validity

The correlations between each of the measures are presented in Table 7. All versions of the downward-extension of the GTS demonstrated large, positive, significant correlations with each other. The GTS-C demonstrated some evidence for convergent validity with a medium, positive correlation with prosocial behaviour. However, the GTS-C did not correlate with loneliness as hypothesised. Peer trust and loneliness were moderately and negatively associated, as demonstrated by the correlation between the GPTS-C and LSDS. The GPTS-C correlated positively with the CGTBS as hypothesised, while the GTS-C did not.

**Table 3. Items for the final version of the General Trust Scale for Children (GTS-C).**

|  | Disagree a lot | Disagree a bit | In Between/Not sure | Agree a bit | Agree a lot |
|---|---|---|---|---|---|
| I am a trusting person | ○ | ○ | ○ | ○ | ○ |
| I think most people tell the truth | ○ | ○ | ○ | ○ | ○ |
| I think most people are good and kind to others | ○ | ○ | ○ | ○ | ○ |
| I think most people trust others | ○ | ○ | ○ | ○ | ○ |
| I can trust most people I meet | ○ | ○ | ○ | ○ | ○ |

*Note.* The instruction for the first item was "*For each sentence on the left, click on how much you agree. There are no right answers, and this is not a test. Just click on the answer that you think fits best for you. First, we want to know more about how you would describe yourself. How much is this sentence true for you:*". For the remaining four items, the instruction was "*Next, we have some questions about what you think about other people.*".

**Table 4. Items for the final version of the General Peer Trust Scale for Children (GPTS-C).**

|  | Disagree a lot | Disagree a bit | In Between/Not sure | Agree a bit | Agree a lot |
|---|---|---|---|---|---|
| Children in my year tell the truth | ○ | ○ | ○ | ○ | ○ |
| Children in my year are good and kind to others | ○ | ○ | ○ | ○ | ○ |
| Children in my year trust other people | ○ | ○ | ○ | ○ | ○ |
| I mostly trust children in my year | ○ | ○ | ○ | ○ | ○ |

*Note.* The instruction for these items was "*Now we would like you to think about the children in your year.*"

**Table 5. Items for the final version of the General Online Trust Scale for Children (GOTS-C).**

|  | Disagree a lot | Disagree a bit | In Between/Not sure | Agree a bit | Agree a lot |
|---|---|---|---|---|---|
| Most people online tell the truth | ○ | ○ | ○ | ○ | ○ |
| Most people online are good and kind to others | ○ | ○ | ○ | ○ | ○ |
| Most people online trust others | ○ | ○ | ○ | ○ | ○ |
| I trust most people online | ○ | ○ | ○ | ○ | ○ |

*Note.* Prior to presenting this scale, children were asked "*Have you ever talked to people online who you do not know in real life, such as in games or comments*" and only presented this scale if selected "*Yes*".

**Table 6. Descriptive Statistics for the Measures of Trust in Others and Psychosocial Outcomes.**

|  | *M* (SD) | Minimum | Maximum | McDonald's ω |
|---|---|---|---|---|
| GTS-C | 18.15 (3.27) | 11.00 | 24.00 | .68 |
| GPTS-C | 14.55 (3.06) | 7.00 | 20.00 | .75 |
| GOTS-C | 10.32 (3.42) | 4.00 | 18.00 | .80 |
| CGTBS | 83.59 (14.34) | 46.00 | 114.00 | .88 |
| PBS | 21.21 (2.28) | 16.00 | 24.00 | .81 |
| LSDS | 8.11 (3.11) | 4.00 | 18.00 | .84 |

*Note.* GTS-C = General Trust Scale for Children. GPTS-C = General Peer Trust Scale for Children. GOTS-C = General Online Trust Scale for Children. CGTBS = Children's Generalized Trust Beliefs Scale. PBS = Prosocial Behaviour. LSDS = Loneliness and Social Dissatisfaction Scale.

**Table 7. Pattern of one-tailed pearson's correlations between measures of trust and psychosocial outcomes.**

|  | 1 | 2 | 3[a] | 4 | 5 | Skewness (S.E.) | Kurtosis (S.E.) | *M* (SD) |
|---|---|---|---|---|---|---|---|---|
| 1. GTS-C | — |  |  |  |  | -.16 (.25) | -.62 (.50) | 18.15 (3.27) |
| 2. GPTS-C | .59*** |  |  |  |  | -.34 (.25) | -.58 (.50) | 14.55 (3.06) |
| 3. GOTS-C[a] | .51*** | .55*** | — |  |  | .37 (.39) | -.14 (.76) | 10.32 (3.42) |
| 4. CGTBS | .15 | .21* | .23 | — |  | -.42 (.25) | -.12 (.50) | 83.59 (14.34) |
| 5. PBS | .26**,[b] | .26**,[b] | .00[b] | .27**,[b] | — | -.40 (.25) | -1.15 (.50) | 21.21 (2.28) |
| 6. LSDS | -.13[b] | -.30**,[b] | -.07[b] | -.23*,[b] | -.25**,[b] | .77 (.25) | .56 (.50) | 8.11 (3.11) |

[a]*n* = 37

[b]Spearman's Rho Correlation

*p < .05.

**p < .01.

***p < .001.

*Note.* GTS-C = General Trust Scale for Children. GPTS-C = General Peer Trust Scale for Children. GOTS-C = General Online Trust Scale for Children. CGTBS = Children's Generalized Trust Beliefs Scale. PBS = Prosocial Behaviour. LSDS = Loneliness and Social Dissatisfaction Scale.

## Discussion

Contrary to our hypothesis, the GTS-C did not correlate with the CGTBS, providing no support for convergent validity with the existing child trust measure. However, some evidence for convergent validity was demonstrated by the correlation between the GTS-C and prosocial behaviour. This finding is consistent with research that has demonstrated an association between general trust and prosociality [17,65]. The lack of association between the GTS-C and the CGTBS may therefore suggest these measures capture different aspects of trust in others. Indeed, while the CGTBS aims to measure generalised beliefs and so might be expected to correlate with the GTS-C, its items exclusively refer to familiar and specific targets [16], and so, a composite score of such beliefs in various known others might be different to a single belief about others in general. Nonetheless, considering that the adult GTS is a well-established measure of general trust, and the predicted association with prosocial behaviour was found, further investigation of the GTS-C is warranted.

Although the reliability of the GTS-C was low, the modified versions of the GTS-C designed to assess general trust in peers and in online contexts were reliable measures of these respective targets. These measures also demonstrated strong correlations with the GTS-C itself. As there is little research into children's online trust beliefs, we did not hypothesise any associations with the GOTS-C. Of note, however, GOTS-C scores appeared lower than trust in peers, indicating children may be more discerning when interacting with strangers online. As only a smaller sample completed this measure, future research would benefit from examining differences in trust in these targets directly. The GPTS-C demonstrated the hypothesised correlations with the CGTBS and loneliness, indicating evidence for its convergent validity and utility separate from the GTS-C. In contrast with previous studies showing no association with peer trust [20], including as measured by the CGTBS peer subscale [8], the GPTS-C demonstrated a similar correlation with prosocial behaviour to the GTS-C. This finding may suggest that – as with the GTS-C – the GPTS-C measures a different aspect of peer trust than that measured than by the CGTBS peer subscale. The CGTBS measure of peer trust describes friends and classmates, whereas the GPTS-C targets all same-age peers. Therefore, the GPTS-C is likely capturing *general peer trust* – more closely related to, though still demonstrably separable from, general trust. The unique patterns of association of the measures of general trust and general peer trust were further examined in Study 3.

## Study 3

Study 3 aimed to address the conflicting literature on age-related differences in general trust and examine the unique associations between general and general peer trust with psychosocial indicators. It also aimed to further examine the psychometric properties of the downward-extended GTS-C and its peer and online versions. Considering the inconsistent associations between these downward-extended scales and existing trust measures in Study 2, parent-rated estimates of general trust, general peer trust, and child trust in online contexts were administered to examine the convergent validity of the three scales. Age-related differences in children's trust in general others, in peers, and online, were examined using the GTS-C, GPTS-C and the GOTS-C. Finally, we investigated the unique relationships of general and general peer trust with friendship and psychosocial indicators to advance understanding of the importance of trust for psychosocial well-being in childhood. This will also provide insight into any differences between general and general peer trust in middle childhood, informing ways in which these constructs are similar and in which they are distinct.

To examine evidence for the internal structure validity of each scale, three CFAs were performed. It was hypothesised that the GTS-C (and similarly, the GPTS-C and the GOTS-C) would demonstrate a good fit for a unidimensional model as originally proposed for the adult GTS [10] and implied by these items all loading on one of the two factors of the IGTS [34]. In support of convergent validity, we hypothesised that parent-rated estimates of child trust in general, in peers, and online, would positively correlate with the GTS-C, GPTS-C, and GOTS-C, respectively. To determine the extent to which general trust and general peer trust may be separable, we also analysed the peer target subscale of the CGTBS. We hypothesised it would correlate positively with the GPTS-C, and that due to the pattern of associations in Study 2, the effect would be small. We examined the correlations between the GTS-C and the CGTBS, and of both these and the peer

trust scales with a measure of loneliness (LSDS). Finally, considering the previously demonstrated associations between prosocial behaviour and general [16,17] but not peer trust [8,20], we hypothesised that parent-rated prosocial behaviour would be significantly correlated with general trust scores measured by the GTS-C.

Prior research leads to conflicting suggestions concerning age-related differences in general trust. However, considering the age-related decreases among younger children in trusting the testimony of non-experts [21,24] and increased critical thinking demonstrated in middle childhood [27], we hypothesised that GTS-C scores would decrease with age. Further, considering the correlations between the measures in Study 2, we examined whether scores on the GPTS-C and GOTS-C also exhibited age-related differences.

Considering the strong, positive correlations between general trust and peer trust found in Study 2, and the quadratic relationships between peer trust with peer victimisation [8] and social acceptance [7], we hypothesised that friendship quality and peer exclusion would have a quadratic relationship with both general and peer trust. Specifically, we predicted a curvilinear association between both the GTS-C and GPTS-C with subscales of child acceptance, child rejection, and the total score on a measure of friendship quality and social skills. Second, based on the quadratic relationship between peer trust and internalised maladjustment, and with poor social preference [7], we hypothesised that peer trust would demonstrate a quadratic relationship with emotional maladjustment and self-perceived social competence, such that measures of these constructs would have a curvilinear association with the GPTS-C.

## Methods

### Participants

Australian children aged 8–12 years and their parents were recruited to complete this study online through social media posts and targeted online advertisements. 208 children aged 8;0–12;11 ($M = 10;4$, $SD = 1.30$) participated, of whom 106 (51.0%) were female and 102 (49.0%) male. Of these, 111 (45% female) indicated they had interacted with strangers online and were included in the online trust group. Overall, 127 (61.1%) of the children were described by their parents as Caucasian, 33 as Asian, 25 as Mixed/Multiracial, 11 as Aboriginal and/or Torres Strait Islander, 5 as Latino and/or Hispanic, 2 as Middle Eastern, 1 as African, 2 as other, 1 as both Caucasian and Aboriginal, and 1 preferring not to say. Parents of the child participants reported their financial circumstances as comfortable ($n = 94$), tight ($n = 87$), and struggling ($n = 27$), with 167 (80.3%) indicating they lived in a household that had been able to meet all essential household expenses within the last year. Participants were recruited between 26 October 2021 and 30 November 2022 and were offered to enter a prize drawer for one of ten AUD$20 vouchers. The sample size recruited was determined by the minimum of 190 required for conducting CFA on unidimensional models with factor loadings of.50 [66].

### Measures

**Parent-report measures.** Parents provided demographic information and were administered three single-item general trust estimates assessing their child's general level of trust overall, in peers, and when communicating online. These parent-report estimates were used to assess the validity of the GTS-C, GPTS-C, and GOTS-C. Parents were asked "*Please indicate* **how you feel your child is generally,** *on scale of 1 (Strongly disagree) to 5 (Strongly agree):* 1. My child is a trusting person. 2. My child is trusting of other children in their year group. 3. My child trusts people they talk to online." There was an additional response point ("Don't know"), and a seventh for the online item ("Not applicable; my child does not talk to others online").

To examine validity, the Friendships and Social Skills Test [67] was administered to assess the quality and quantity of children's friendships and social well-being. The 25 items (rated on a four-point scale from "Never or does not apply" to "Almost always") have good internal consistency ($\alpha = .89$) and assess subscales of expressed concern (regarding the level of parental concern over the child's social functioning), prosocial skills, negative social behaviours, child acceptance (by

other peers), and child rejection (by other peers). The prosocial skills and child acceptance subscales are reverse scored, such that higher scores on each subscale and overall indicates higher parental concern over the child's social functioning.

Behavioural competencies and difficulties were assessed with the Child Adjustment and Parent Efficacy Scale (CAPES; α = .90) [68]. This 27-item scale asks parents to rate how true each of the described behaviours have been for their child over the past four weeks, on a four-point scale from 0 ("Not at all") to 3 ("Very much"). In addition to a total intensity score measuring overall child adjustment, the CAPES assesses subscales of behavioural intensity (α = .90) and emotional maladjustment (α = .74) [68].

### Child self-report measures

To assess general trust beliefs, the downward-extended five-item GTS-C, as well as the four-item peer (GPTS-C) and online (GOTS-C) versions as described in Study 2, were administered to children.

To examine validity, the CGTBS [16], as described in Study 2, was administered to assess convergent validity for the developed trust scales. In addition to this total score, the six items assessing trust in peers have demonstrated utility as a peer trust subscale (CGTBS Peer) with adequate internal consistency (α = .79) [8]. Responses to this subscale were also included for analysis.

For the regression analyses, psychosocial well-being was assessed with the six-item Self-Perceived Social Competence (SPSC) subscale of the Self-Perception Profile for Children – Grades 3–8 [69]. This measure assesses the extent to which children feel successful in social settings and has good internal consistency (α = .78—.90). Participants indicate which of two dichotomous self-perceptions best describe them, then indicate the extent to which this perception is true of them (out of "Really true for me" and "Sort of true for me", such that their social competence is estimated on a four-point scale for each item.

Additionally, the four-item LSDS [59] was administered as described in Study 2.

### Procedure

Parents completed the survey online via Qualtrics, first providing informed consent for their own and for their child's participation, and confirming eligibility (by providing their child's age and confirming they were living in Australia). They then provided demographic information, and completed the measures described above. Following this, they were invited to indicate whether they would be interested in participating in a one-year longitudinal follow-up, and to ask their children to participate in the remainder of the survey by clicking on a link directing them to the children's participant information form. This form described the research aims and procedure in plain language and indicated what to do if the child did not wish to continue at any time. Children then provided assent by clicking on 'Yes' when asked whether they wanted to participate in the research.

### Analytical procedure

Data were screened and checked for missing values, with any missing values analysed for randomness using Little's missing completely at random (MCAR) test.Influential cases were examined using the interquartile range rule method [60], and normality examined by calculating the ratio of skewness and kurtosis to each of its respective standard errors. Internal structure validity was examined by conducting a CFA on a unidimensional model of the GTS-C, GPTS-C, and GOTS-C, confirmed by a non-significant chi-square test for exact fit and examining the fit indices. For each, a CFA was conducted on the unidimensional model proposed for the original adult GTS [10] using full maximum likelihood estimation. The absolute goodness-of-fit of the model was evaluated using the chi-square test for exact fit and by examining the comparative fit index (CFI), the Tucker-Lewis Index (TLI), the root mean square error of approximation (RMSEA) with 90% confidence interval (CI), and the standardised root mean square residual (SRMR). Indices indicating good model fit include a

non-significant chi-square value, and high CFI and TLI values – above.90 are considered acceptable and above.95 very good [70]. Values of less than.06 are considered ideal for the RMSEA and SRMR [70], although as high as.08 are also considered reasonable [71].

Age-related differences in general, peer, and online trust were examined by conducting linear trend analyses on the GTS-C, GPTS-C, and GOTS-C across the five age groups in years. Reliability and convergent/discriminant validity analyses were conducted using a similar strategy as in Study 2, though due to some of the non-significant results in Study 2 that were contrary to predictions, two-tailed correlations were conducted here. Additionally, curvilinear regression analyses were conducted to examine whether a quadratic or linear model was a better fit for the relationship between both general trust and peer trust with psychosocial well-being outcomes. The analyses were conducted using SPSS 27.0.1, except for conducting CFA using jamovi [63].

## Results

### Preliminary analysis

Data were initially screened to ensure that parent responses were available for all completed child data. Responses were included for analysis when both the parent participant had completed the survey and opened the child portion of the survey (such that all data included parent-child response pairs), and where the child participant had not dropped out of the survey before completing all trust measures and at least one of the measures of psychosocial well-being.

Data were screened for missing values prior to conducting analysis, revealing that parent-report values were missing for the CAPES and FASST, and child-report values were missing for the CGTBS and SPSC scales. Additionally, two child participants dropped out before completing the SPSC and were not included for analysis involving this measure. All other data from these participants were included for analysis. A series of Little's tests indicated that values were MCAR for all scales except the CAPES Emotional Maladjustment subscale ($\chi^2(6) = 13.43$, $p = .037$).

For this subscale, a dummy variable was coded indicating whether a missing value was present. A series of independent-samples t-tests were conducted on the psychosocial well-being variables comparing groups on the dummy variables to determine whether the values were missing at random (MAR). These t-tests indicated that missing values on this subscale may have been explained by variance in parent-report estimates of children's negative social behaviours ($t(206) = -2.17$, $p = .03$), indicating the data were likely MAR. As such, expectation-maximisation imputation was used to impute eight missing values on the SPSC, one on the CGTBS, one on the FASST, 11 on the CAPES Behavioural Problems subscale, and 3 values on the Emotional Maladjustment subscale.

After missing values were imputed, variables were screened for influential cases using the interquartile range process, such that values more than 2.2 times greater than the third quartile or smaller than the first quartile were deemed influential [60]. This process identified one influential case above the cut-off for the Child Rejection, one for the Expressed Concern and ten cases for the Negative Social Behaviours subscale of the FASST, as well as one for the FASST total score, and two influential cases below the cut-off for the GTS-C. The values identified were replaced with the value 2.2 times the upper or lower quartile on that variable, respectively.

The ratios of skewness and kurtosis to their respective standard error scores were calculated for each variable of interest. Scores on the LSDS, the Expressed Concern, Child Rejection, and Negative Social Behaviours subscales of the FASST and total FASST, and the Emotional Maladjustment subscale of the CAPES violated the assumption of normality, as demonstrated by ratios larger than |2.58|– the cut-off recommended for samples greater than 200 [64]. As such, these variables were treated as non-parametric for correlational analyses. All other variables had ratios below the cut-off and were thus determined to be normally distributed. An examination of the histograms for each variable also confirmed this conclusion. As such, the data were deemed appropriate for further analysis.

Descriptive statistics by age group are presented in Table 8 for the measures of trust. Descriptive statistics for all other outcome measures are presented in Table 11.

**Table 8. Mean Scores (SD) on trust scales and subscales, by age group.**

| Variable | Age | | | | |
|---|---|---|---|---|---|
| | 8 (*n* = 32) | 9 (*n* = 52) | 10 (*n* = 52) | 11 (*n* = 42) | 12 (*n* = 30) |
| GTS-C | 18.81 (2.61) | 18.96 (3.29) | 18.87 (3.00) | 18.48 (3.57) | 17.01 (3.41) |
| GPTS-C | 15.06 (2.68) | 15.81 (2.58) | 14.15 (2.85) | 14.6 (3.33) | 12.87 (3.36) |
| GOTS-C | 12.5 (3.03) | 11.89 (3.36) | 11.97 (3.49) | 11.7 (4.05) | 10.28 (3.43) |
| CGTBS Total | 81.35 (12.79) | 84.58 (15.34) | 85.29 (13.57) | 85.67 (15.73) | 83.9 (12.98) |
| CGTBS Peer | 20.47 (3.89) | 20.88 (4.46) | 21.14 (3.89) | 20.71 (5.14) | 21.23 (4.63) |
| Trust in General[a] | 4.37 (.98) | 4.31 (.73) | 4.40 (.64) | 4.10 (.97) | 4.30 (.75) |
| Trust in Peers[a] | 4.47 (.88) | 4.19 (.89) | 4.20 (.73) | 4.24 (.73) | 3.83 (1.07) |
| Trust Online[a] | 4.22 (.67) | 3.22 (1.19) | 3.50 (1.20) | 3.70 (1.12) | 2.95 (1.16) |

[a]Single-item parent-report measure

*Note*: for Trust Online (parent-report estimate), *n*s for each group were 9, 41, 40, 30, and 21, respectively. GTS-C = General Trust Scale for Children. GPTS-C = General Peer Trust Scale for Children. GOTS-C = General Online Trust Scale for Children. CGTBS Total = Children's Generalized Trust Beliefs Scale total score. CGTBS Peer = Children's Generalized Trust Beliefs Scale peer subscale score.

**Table 9. Pattern of two-tailed pearson's correlations between measures of trust.**

| | 1 | 2 | 3 | 4 | 5 | 6 | 7 | *M (SD)* | ω | Skewness (*SE*) | Kurtosis (*SE*) |
|---|---|---|---|---|---|---|---|---|---|---|---|
| 1. GTS-C | — | | | | | | | 18.54 (3.24) | .70 | -.13 (.17) | -.21 (.34) |
| 2. GPTS-C | .56*** | — | | | | | | 14.61 (3.06) | .77 | -.29 (.17) | -.47 (.34) |
| 3. GOTS-C | .44*** | .43*** | — | | | | | 11.67 (3.50) | .82 | .15 (.23) | -.33 (.46) |
| 4. CGTBS | .33*** | .30*** | .27*** | — | | | | 84.41 (41.21) | .85 | .18 (.17) | -.33 (.34) |
| 5. CGTBS Peer | .36*** | .36*** | .31*** | .78*** | — | | | 20.91 (4.38) | .68 | -.07 (.17) | -.10 (.34) |
| 6. Trust in General[a] | .14*,b | .19**,b | .10b | .04b | .12b | — | | 4.30 (.81) | — | — | — |
| 7. Trust in Peers[a] | .24***,b | .32***,b | .07b | .04b | .06b | .40***,b | — | 4.20 (.86) | — | — | — |
| 8. Trust Online[a] | -.03b | -.03b | .32***,b | -.11b | -.03b | .08b | .13b | 3.43 (1.18) | — | — | — |

[a]Parent-report single-item estimate

[b]Spearman's Rho Correlation

*p < .05.

**p < .01.

***p < .001

## Reliability

All versions of the GTS-C demonstrated acceptable internal consistency (see Table 9).

## Internal structure validity

To establish evidence for internal structure, the factor structures of the modified trust scales (GTS-C, GPTS-C, GOTS-C) were examined through CFA. For the GTS-C, fit indices for the unidimensional model demonstrated a non-significant test for exact fit, with all other indices also meeting the criteria for a very good fit ($\chi^2(5) = 2.90$, $p = .72$, CFI = 1.00, TLI = 1.03, SRMR = .02, RMSEA < .001, 95% CI [.00,.07]). All five items loaded onto the general factor with standardised estimates above or equal to .35 (see Table 10). The GPTS-C demonstrated a significant chi-square test, though most other indices met criteria for a very good fit ($\chi^2(2) = 6.71$, $p = .04$, CFI = 0.98, TLI = 0.93, SRMR = .03, RMSEA = .11, 95% CI [.02,.20]). Similarly, the GOTS-C demonstrated a significant chi-square test, but good model fit indices for the CFI and SRMR ($\chi^2(2) = 7.74$, $p = .02$, CFI = 0.96, TLI = 0.89, SRMR = .03, RMSEA = .16, 95% CI [.05,.29]). These analyses provided evidence for

**Table 10. Factor Loadings of the GTS-C, GPTS-C, and GOTS-C.**

| Item | GTS-C | GPTS-C | GOTS-C |
|---|---|---|---|
| 1. I think [target] tell the truth | .35 | .76 | .85 |
| 2. I think [target] are good and kind to others | .52 | .68 | .64 |
| 3. I think [target] trust others | .69 | .58 | .68 |
| 4. I can trust [target] I meet | .68 | .66 | .74 |
| 5. I am a trusting person | .55 | — | — |

the internal structure validity of the single-factor, five-item GTS-C, as well as providing support for a unidimensional structure for the four-item GPTS-C and GOTS-C.

## Correlational analyses

Table 9 displays the results of the correlational analyses examining the associations between the measures of trust in others. Evidence for convergent validity of the GTS-C was demonstrated with the positive, large correlation with the CGTBS. For the GPTS-C, the positive, large correlation with the Peer subscale of the CGTBS also provided evidence of convergent validity. As hypothesised, the parent-report estimates of child trust were also positively correlated with each respective scale, though these associations were stronger for the GPTS-C than the GTS-C.

The associations between the GTS-C and GPTS-C with measures of friendship and psychosocial well-being outcomes were analysed through a series of correlational analyses (see Table 11). Convergent validity was demonstrated for the GTS-C with medium correlations in the hypothesised directions for social competence and total friendship quality scores on the FASST. The GTS-C was also negatively but weakly to moderately associated with the measure of loneliness, and the child rejection, expressed concern, and negative social behaviours subscales of the FASST, while the GPTS-C was moderately to strongly and negatively associated with all variables except for negative social behaviours, which was not significantly correlated. The CAPES Emotional Maladjustment subscale also demonstrated a small, negative association with the GTS-C, though no other correlations were statistically significant. Stronger evidence for convergent validity was demonstrated for the GPTS-C, for which all correlations were significant and in the hypothesised direction, except for the association with prosocial skills.

Finally, a contrast analysis on the GOTS-C scores revealed no significant difference in mean trust scores online across the age groups ($t(106) = 1.57$, $p = .12$).

## Curvilinear regression analyses

Curvilinear regression was conducted on the measures of psychosocial well-being predicted by the GTS-C and GPTS-C scores, respectively. The GTS-C did not demonstrate the hypothesised quadratic relationship with child acceptance ($\beta = -.41$, $\Delta R^2 < .01$, $p = .50$), nor with child rejection ($\beta = .90$, $\Delta R^2 = .01$, $p = .13$), or total FASST scores ($\beta = .20$, $\Delta R^2 < .01$, $p = .74$). The GPTS-C also demonstrated no significant quadratic effects with child acceptance ($\beta = .74$, $\Delta R^2 < .01$, $p = .17$), child rejection ($\beta = .14$, $\Delta R^2 < .001$, $p = .79$), or total FASST scores ($\beta = -.27$, $\Delta R^2 < .01$, $p = .60$). Similarly, no significant quadratic effects were demonstrated between the GPTS-C and SPSC ($\beta = .18$, $\Delta R^2 < .01$, $p = .73$) or emotional maladjustment CAPES subscale scores ($\beta = .02$, $\Delta R^2 < .01$, $p = .97$), respectively.

## Discussion

The GTS-C and its peer and online versions all demonstrated evidence of reliability and validity across the series of analyses. Evidence for internal structure validity was demonstrated, with the unidimensional model of the GTS-C demonstrating excellent model fit. Additionally, the parent-rated estimates of child trust demonstrated convergent validity for

**Table 11.  Pattern of two-tailed pearson's correlations between trust and psychosocial well-being variables.**

| Variable | 1 | 2 | M (SD) | ω | Skewness (S.E.) | Kurtosis (S.E.) |
|---|---|---|---|---|---|---|
| 1. GTS-C | — | | 18.54 (3.24) | .70 | -.13 (.17) | -.21 (.34) |
| 2. GPTS-C | .56*** | — | 14.61 (3.06) | .77 | -.29 (.17) | -.47 (.34) |
| 3. Social Competence | .25*** | .25*** | 2.74 (.64) | .58 | -.42 (.17) | -.60 (.34) |
| 4. Loneliness | -.15*,a | -.28***,a | 8.63 (3.60) | .87 | .59 (.17) | -.12 (.34) |
| 5. Child Acceptance | -.05 | -.22*** | 8.24 (3.30) | .81 | -.31 (.17) | .15 (.34) |
| 6. Child Rejection | -.21***,a | -.37***,a | 3.41 (3.49) | .89 | .96 (.17) | .18 (.34) |
| 7. Expressed Concern | -.23***,a | -.38***,a | 3.29 (3.66) | .90 | 1.04 (.17) | .14 (.34) |
| 8. Negative Social Behaviours | -.15*,a | -.10a | 2.90 (2.73) | .84 | .99 (.17) | .10 (.34) |
| 9. Prosocial Skills | -.05 | -.07 | 5.43 (2.96) | .71 | .30 (.17) | -.63 (.34) |
| 10. FASST Total | -.20***,a | -.37***,a | 23.35 (11.09) | .87 | .48 (.17) | -.33 (.34) |
| 11. Emotional Maladjustment | -.20***,a | -.27***,a | 2.92 (1.99) | .71 | .62 (.17) | -.27 (.34) |
| 12. Behavioural Problems | -.11 | -.19** | 22.04 (11.28) | .91 | .29 (.17) | -.78 (.34) |
| 13. CAPES Total Intensity | -.12 | -.21*** | 24.96 (12.49) | .91 | .32 (.17) | -.69 (.34) |

aSpearman's Rho Correlation

*$p < .05$.

**$p < .01$.

***$p < .001$.

Age-related differences in trust beliefs

A contrast analysis was conducted on mean GTS-C scores to test whether generalised trust scores would exhibit a negative linear effect across the five age groups (using weights of 2, 1, 0, -1, and -2). The linear contrast was significant ($t(203) = 2.32$, $p = .021$, $r^2_{effect} = .026$), indicating as hypothesised that GTS-C scores were lower in the older children. However, the small effect size suggested that less than 3% of the variance in the five GTS-C mean scores was accounted for by the linear trend.

A similar contrast analysis was conducted on GPTS-C scores, revealing a significant negative linear trend in children's trust in peers ($t(203) = 3.47$, $p < .001$, $r^2_{effect} = .054$), suggesting 5.4% of the variance in GPTS-C scores was accounted for by the effect of age-related differences.

each of the GTS-C, GPTS-C, and GOTS-C. The magnitude of these correlations was not especially strong for convergent validity, though this finding was not surprising given low agreement between parent-report and child self-report estimates in other domains [72]. The GTS-C and GPTS-C also demonstrated convergent validity with the CGTBS and its peer target subscale, respectively, unlike in Study 2. The hypothesised correlations with loneliness that were absent in Study 2 were evident in this larger sample, though the medium correlation with peer trust suggested stronger convergent validity for the GPTS-C than the GTS-C. The hypothesis that general trust and prosocial behaviour would be positively correlated was not supported. Both general and peer trust demonstrated small but significant age-related decreases, as hypothesised. Despite a similar age cohort as in previous research [7,16,], none of the hypotheses concerning quadratic relationships between trust and psychosocial indicators were supported.

The support for convergent validity with the CGTBS suggests that perhaps the null finding in Study 2 were due to the smaller sample size in that study. It is worth noting that the correlations with the CGBTS and its peer subscale were still smaller in this larger sample than usually seen in measures of the same construct. This smaller association could suggest that general trust, as measured by the GTS [10], is not conceptually identical to trust beliefs generalised across a range of targets. This finding is important considering the longstanding disconnect between how child and adult trust has been conceptualised and measured [18]. The creation of a general trust scale for children, downward extended from an established adult measure, takes the first step in addressing this problem and will enable investigations of how and even whether this construct develops through childhood and into adulthood.

In this study, the demonstrated age-related differences suggest that general trust decreases slightly through middle childhood. This finding aligns with research into age-related differences in children's critical thinking [23,27], increased

discernment during trust in testimony [22,24], and selective trust [25,26], suggesting there may be a general trend between early and middle childhood of becoming more discerning of information during social interactions. It may also imply that findings of age-related increases in trust behaviours [28] do not reflect beliefs or attitudes toward others more broadly. Additionally, the age-related differences in both peer and general trust scales implies that trust in others is more likely affected by social experience than a stable disposition. However, the effect sizes for the linear trend in both scales accounted for only a small proportion of variance in mean scores. This small effect may indicate that by middle childhood, most children have developed their level of trust beliefs in others. Additionally, confidence in this finding is limited by our inability to assess measurement invariance between the age groups. Future research would benefit from recruiting a large enough sample to conduct multigroup CFA, and examining whether age-related decreases in general and peer trust beliefs are more pronounced during earlier phases of development.

The lack of association between measures of trust and prosocial behaviour is in line with other null findings in the literature [8,20], but contradicts those that have demonstrated an association [16,17]. This finding may suggest that neither trust in general, nor in peers, influences how willing a child may be to engage in behaviours that benefit others. However, this finding was surprising considering the significant correlations demonstrated with the same trust scales in Study 2. In this study, prosocial behaviour was measured with a parent-report scale. Previous research has demonstrated that parent-report and child self-report measures can demonstrate low cross-informant correspondence [73]. Additionally, adult research using the GTS has demonstrated an association with prosociality [34], with a similar strength to that demonstrated in Study 2, suggesting there may be some relationship. To determine whether these constructs are indeed related in childhood, a larger study would benefit from administering measures of both prosociality and general trust to the same rater.

Overall, the psychosocial correlates were associated more strongly with peer than general trust. This finding appears in line with much of the previous literature that has demonstrated associations between various measures of trust in peers with other psychosocial indicators [1,4,8,15]. While the correlations hypothesised to support convergent validity for both the GTS-C and GPTS-C were all significant (except for with prosocial behaviour), the GPTS-C was uniquely associated with child acceptance, behavioural problems, and overall child adjustment. Considering the breadth of these social difficulties and the stronger correlations with friendship outcomes than the GTS-C, measures of peer trust may be more insightful than general trust for outcomes relevant to children's social functioning.

The results of the quadratic analyses contradict previous research suggesting both high- and low-trusting children are at risk of poor social acceptance [7]. While it could be argued this finding is due to the measurement of different aspects of social functioning such as friendship quality, the subscale scores for both child acceptance and child rejection assess social acceptance, neither of which were significant. As stated above, the methodology used in this study was limited by measuring most psychosocial correlates with parent-report measures while estimating both general and peer trust with child self-report measures. However, the examination of this relationship with social competence, measured with the child-report SPSC, was not limited by this approach. The absence of a quadratic relationship between general or peer trust and social competence combined with the significant positive correlations suggests a linear relationship better explains at least this association.

## General discussion

This report provides a comprehensive exploration of the outcomes of general trust in middle childhood. Over three studies, this research has demonstrated the unique patterns of association with psychosocial correlates of general trust and general peer trust, indicating these constructs are separable. The findings from these studies also demonstrate clear, though small, age-related decreases in general and general peer trust during middle childhood. Studies 1 and 2 demonstrate the appropriateness of measuring trust directly in a general target in middle childhood, and evidence for the validity and reliability of a downward extension of Yamagishi and Yamagishi's [10] GTS, and of modified versions examining peer

and online targets, resulting in three scales measuring child trust beliefs. Study 3 provides evidence that while both general and general peer trust are associated with psychosocial well-being, general peer trust is more closely associated with a broader range of social difficulties.

A central finding of this research was the confirmation that, within the middle childhood stage, older children exhibit less general trust than younger children. Sutter and Kocher [28] suggested that trust increases linearly from middle childhood into early adulthood, when it stabilises throughout the rest of the lifespan. Indeed, much research into adult trust in others asserts it is generally stable [33,74–76]. However, the present findings demonstrate that unlike behavioural trust, general trust decreases during childhood and is likely a separable construct from the behaviour measured by the trust game that may measure more prosocial or economic behaviours [31]. This finding aligns more closely with research into child trust beliefs that demonstrate older children exhibit less trust in inconsistent or domain-irrelevant testimony during early childhood than younger peers [21,24]. This decrease may also align with other findings from the developmental literature, suggesting that general trust may develop inversely with critical stance, thought to increase through adolescence into adulthood [27]. The current research is the first to examine trust beliefs in middle childhood and suggests children may become more discerning with age through this developmental period. Interestingly, general peer trust exhibited stronger decreases still. This result may indicate support for theoretical conceptualisations that assert social interaction directly impacts trust development [77], assuming children spend more time interacting with others their own age. However, both effect sizes were relatively small, so the effect of social interaction on levels of these forms of trust would need to be examined directly. In either case, the present findings provide further evidence that any stability in adults seems likely to follow a period of development in childhood. Future research is needed to confirm whether longitudinal or age-related changes are present in older samples.

These studies demonstrate the utility of the developed measures of trust beliefs for children. Studies 1a and 1b together suggest that children can understand the word 'trust' in ways that are consistent with adults, and thus are able to respond to these items asking directly about their trust in a valid way. This understanding is significant as it demonstrates that a brief measure of trust in others is appropriate for children of this age. Prior approaches to measuring children's trust in others have involved time-consuming behavioural tasks [28,78] and longer questionnaires with lengthy items [16,40]. A brief and valid measure such as the GTS-C or GPTS-C will be invaluable for future research into this important construct. Further, the unique pattern of associations demonstrated by the GOTS-C suggests it measures a distinct construct and will be a useful tool for research into trust online. Future research will be able to investigate the impact of individual differences in child trust in digital contexts compared with how children trust face-to-face, which may have important implications for risk-aversion and safety online.

While the GTS-C exhibited convergent validity with the CGTBS in Study 3, this association was not uniform across studies. This inconsistency suggests potential conceptual distinctions between general trust measured by the GTS-C and the trust beliefs measured by the CGTBS directed at familiar and specific targets. While at first this hinted at poor convergent validity, it is possible that the GTS-C, GPTS-C, and CGTBS each measure related but separable constructs. The GTS-C and GPTS-C demonstrated unique patterns of association with psychosocial correlates previously shown to relate to general trust in Study 3, suggesting that the measures are valid, and providing evidence for the separability of general trust and the concept of general peer trust. The consistent associations across the studies between the CGTBS and GPTS-C, measuring the more familiar target of peers, further supports the notion that the CGTBS measures a general trust belief in familiar targets, rather than general trust as conceptualised in adult trust research [34,35,39]. This distinction implies a novel theoretical aspect of trust beliefs in others, that general peer trust is separable from but related to both general trust and those beliefs measured by the CGTBS. Such a distinction was demonstrated in the unique pattern of associations among the trust measures but is not surprising considering the differences in the degree of familiarity and specificity in the target each construct aims to measure. However, as the only unique association with the GTS-C was with negative social behaviour, future research should confirm whether the GPTS-C is a superior measure or whether the GTS-C uniquely relates to other important developmental outcomes.

Collectively, these studies contribute to the nuanced conceptualisation of trust in middle childhood. General peer trust emerged as a predictor of more psychosocial indicators than general trust and was frequently more strongly associated. The relationships with child acceptance, behavioural problems, and overall child adjustment in particular underscore the relevance of general peer trust in understanding children's social functioning. Together with the results concerning age-related differences in general trust, the findings from the present research may have practical implications for educators, parents, and other practitioners involved in children's development. Understanding the distinct nature of general and peer trust could inform the design of age-appropriate interventions aimed at promoting positive social interactions and trust-building among peers. The age-related decreases in trust may indicate that this period is sensitive, such that the benefits of higher trust levels for better friendship outcomes and less rejection from peers need to be accessed among younger groups before their trust levels begin to decline.

The findings concerning associations with prosocial behaviour were arguably the most inconsistent across the studies. Whereas previous research was limited, it did indicate a clear association with general [17,34,65] and not peer trust [1,8,15,20]. This pattern was supported by the findings in Study 2, whereas the subscale measure of prosociality did not correlate with any of the trust measures in Study 3. The shift from a child self-report measure in Study 2 to parent-report in Study 3 prompts consideration of whether informant discrepancies explain this inconsistency, or whether prosocial behaviours are prone to social desirability bias in self-report measures. Indeed, prior null findings involved teacher-rated scales [20], or behavioural observations of prosocial behaviour [8], rather than self-report questionnaires. Future research may benefit from comparing measures of prosocial behaviour to determine whether informant discrepancies explain these contradictions, and whether any measures relate to general or peer trust.

The current methodology was limited by performing the prompt in Study 1A following the battery of measures administered in Study 2. Participants were not provided any language in the battery that would help define trust, but it does raise the possibility that child participants were able to identify from context that the concept referred to in the GTS-C was relevant to the vignettes of the CGTBS [16] and possibly tried to describe the nature of the decision made in those vignettes when defining trust. However, there was a very large proportion who successfully defined trust, and the findings from Study 1B demonstrated the robustness of this proportion due to the non-significant effect of presentation order in an adult sample. As such, it seems unlikely this presentation order was driving a better conceptual understanding of the term. Considering also previous research that children can understand the word trust by 8 years [51], the presentation order may have at most improved children's recall of prior learning.

## Conclusions

This series of studies navigates the intricate landscape of trust in middle childhood. The current research demonstrated that the GTS-C, GPTS-C, and GOTS-C are valid and reliable measures of trust in others in middle childhood. The impact of general trust on child development has long been a focus of research [18], but existing measures of children's general trust are lengthy or conceptually distinct from conceptualisations of trust used in adult research [16,17]. The modified measures developed in this research improve our understanding of how trust differs among different developmental stages (rather than being stable throughout the lifespan), and how it relates to friendship quality and psychosocial well-being. The GTS-C and its adaptations provide a valuable contribution to our current and future understanding of trust, highlighting target-specific nuances that should be considered in future research. Access to a measure that is both brief and conceptually aligned to adult research will aid in understanding how trust shapes childhood development and help to unify conceptual differences between fields.

## Supporting information

**S1 Appendix. Codebook for process of scoring trust definitions.**
(DOCX)

## Acknowledgments

We would like to thank all children and families who participated in the studies. Moreover, we are grateful for the contributions to the project and help with data collection from Anika Gosling in Studies 1A and 2, Aisling Morgan-McHale in Study 3, and to the UWA Children's Research Project for assistance with recruitment and data collection for all studies reported here.

## Author contributions

**Conceptualization:** Alex Wheeler, Donna M. Bayliss, Jeneva Ohan.

**Data curation:** Alex Wheeler.

**Formal analysis:** Alex Wheeler.

**Investigation:** Alex Wheeler.

**Methodology:** Alex Wheeler, Donna M. Bayliss, Jeneva Ohan.

**Project administration:** Alex Wheeler, Donna M. Bayliss.

**Supervision:** Donna M. Bayliss, Jeneva Ohan.

**Writing – original draft:** Alex Wheeler.

**Writing – review & editing:** Alex Wheeler, Donna M. Bayliss, Jeneva Ohan.

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
