## [Decision Letter · Decision Letter 0]

27 Sep 2024

PONE-D-24-23774Age-related differences in trust beliefs during middle childhood: Downward-extension and validation of the General Trust ScalePLOS ONE

Dear Dr. Wheeler,

Thank you for submitting your manuscript to PLOS ONE. After careful consideration, we feel that it has merit but does not fully meet PLOS ONE’s publication criteria as it currently stands. Therefore, we invite you to submit a revised version of the manuscript that addresses the points raised during the review process. Please focus primarily on feedback from Reviewer 1 (the longer one). With regards to Reviewer 2, the most important observation is that "in Table 5, it would be interesting if the authors could provide a test for comparing means to investigate the presence of statistically significant differences in means according to age."

We look forward to receiving your revised manuscript.

Kind regards,

Frantisek Sudzina

Academic Editor

PLOS ONE

Journal Requirements:

2. For studies involving third-party data, we encourage authors to share any data specific to their analyses that they can legally distribute. PLOS recognizes, however, that authors may be using third-party data they do not have the rights to share. When third-party data cannot be publicly shared, authors must provide all information necessary for interested researchers to apply to gain access to the data. (https://journals.plos.org/plosone/s/data-availability#loc-acceptable-data-access-restrictions) For any third-party data that the authors cannot legally distribute, they should include the following information in their Data Availability Statement upon submission: 1) A description of the data set and the third-party source 2) If applicable, verification of permission to use the data set 3) Confirmation of whether the authors received any special privileges in accessing the data that other researchers would not have 4) All necessary contact information others would need to apply to gain access to the data.

Reviewers' comments:

Reviewer's Responses to Questions

**Comments to the Author**

1. Is the manuscript technically sound, and do the data support the conclusions?

Reviewer #1: Yes

Reviewer #2: Partly

2. Has the statistical analysis been performed appropriately and rigorously? 

Reviewer #1: Yes

Reviewer #2: I Don't Know

3. Have the authors made all data underlying the findings in their manuscript fully available?

Reviewer #1: No

Reviewer #2: No

4. Is the manuscript presented in an intelligible fashion and written in standard English?

Reviewer #1: Yes

Reviewer #2: Yes

5. Review Comments to the Author

Reviewer #1: The article makes a significant contribution to an area of research gap in the socio-emotional life of children.

The paper is well grounded at a theoretical level and provides relevant bibliographical references to support its hypotheses and assumptions.

The three studies are theoretically and methodologically coherent.

In study 1, it would be interesting if the authors could provide textual examples of the participants' responses.

In studies 2 and 3, the authors should explain the criteria they used to determine the sample size.

Finally, in Table 5, it would be interesting if the authors could provide a test for comparing means to investigate the presence of statistically significant differences in means according to age.

The discussion is adequate, although it does not fully justify the need to provide a general measure of confidence based on the results obtained

Reviewer #2: The article presents the development of a scale with a fairly solid design. It uses various samples, which, although they may be subject to criticism, I believe are adequate for the purpose of the research. I think that due to the complexity of the design, the article is difficult to follow, especially the methods and results.

Below, I outline some general guidelines to give coherence to the manuscript, and especially to make it easier to read for those who may not be familiar with scale validations. Personally, I am familiar with them, and I found it difficult to follow the rationale.

In the introduction, not all the analyses/hypotheses that will be carried out are presented. In fact, in Study Three, many things are introduced that were not mentioned beforehand. I believe that the CFA is precisely a strong point of the study, and the authors do not mention it.

In each methods section, the data analyses carried out should be presented. For example, in the more qualitative analysis, various indices or coefficients are mentioned without explaining what they are or how they were calculated. The same happens with the quantitative studies. It is not until the results section that it becomes clear what is going to be done, which makes the text difficult to follow.

After reviewing these general aspects, I will be able to provide more specific feedback on the issues in the manuscript. Below, I mention some specific areas for improvement.

(1) These studies also demonstrated the psychometric properties. ->“These studies also provides evidence of the psychometric quality of the scales. The psychometric properties can not be fully demonstrated (ever). In the same line, I suggest authors do not say “validate” but “provide evidence of validity” across the manuscript.

(2) Construct validity is a concept of the Standards for Psychological and Educational testing (1985) that was updated in 1999. Currently, we have “content validity”, “internal structure” and “ relationship with external variables”. For concision (page 8), if I am not wrong, in the first study you provide test content and response process, in study 2, relationship with external variables (convergent/discriminant). In third study, you provide further evidence of relationship with external variables, in this case based on differences between relevant groups. I wonder if it is possible to provide evidence bases on the internal structure (i.e.,factor analysis).

(3) I think it is necessary to state where this study has been carried (I assume in Australia for the ethics office, but not sure). Did children or families receive any gratification for participation?

(4) I do not understand line 246. I think some verb or noun is missing.

(5) Design section I think is “Data analysis section”. More description is needed in order to make the study reproducible.

(6) I am struggeling understanding Table 1/2. Some extra information about what this table presents (or adding information into a data analysis section). How is reliability measured?

(7) In Study 1B is needed further discussion. As you are using undergraduate students they may have more knowledge that “general adults”.

(8) Authors are generating sum scores without providing evidence of internal structure. This is a problem since we do not have evidence that this can be done. As authors are using omega that depends on CFA, please report the results of CFA.

(9) Table 4. Add means and SD

(10) Here we finally find the CFA. This has not mentioned before, and I think it is necessary to present it. You need to further developed the estimators and correlation matrix used.

(11) “Structural validity” does not exist “internal structure validity evidence”

(12) To assess for age differences, it is necessary to perform a multi-group confirmatory factor analysis before.

(13) Which software was used?

(14) IC of RMSEA is nearly impossible to be 95%: it is probably 90%

(15) I suggest the authors to present chi/df and CFI, TLI, RMSEA in the same parenthesis.

6. PLOS authors have the option to publish the peer review history of their article (what does this mean? ). If published, this will include your full peer review and any attached files.

**Do you want your identity to be public for this peer review?** For information about this choice, including consent withdrawal, please see our Privacy Policy .

Reviewer #1: No

Reviewer #2: No

---

## [Author Response · Author response to Decision Letter 1]

25 Mar 2025

Reviewer #1: The article makes a significant contribution to an area of research gap in the socio-emotional life of children.

The paper is well grounded at a theoretical level and provides relevant bibliographical references to support its hypotheses and assumptions.

The three studies are theoretically and methodologically coherent.

Response: Thank you for this feedback. We would also like to declare that additional revisions have been made to the theoretical justification for these studies in response to feedback on the corresponding author’s PhD dissertation (particularly in the first few pages of the introduction, lines 74-116). Throughout, we refer to the page and line numbers within the track-changes manuscript.

1. In study 1, it would be interesting if the authors could provide textual examples of the participants' responses.

Response: While we agree that textual examples would be preferable, our ethics approval did not include asking permission for individual participant responses to be included in publications, so we are unfortunately not able to offer this inclusion. Instead, we have amended the manuscript to add a codebook as an appendix that includes sample items, many of which were similar in nature to the responses collected.

2. In studies 2 and 3, the authors should explain the criteria they used to determine the sample size.

Response: Thank you for this suggestion. We have added these criteria to the ends of the Participants sections for studies 2 and 3. For example on pg. 20, “A power analysis conducted in G*Power 3.1 (57) determined a minimum sample size of 44 would be required to detect an effect size demonstrated between the CGTBS and a prosocial behaviour in prior research (16) with 80% power and a two-sided significance level of .05. A convenience sample of at least double this requirement was recruited in an attempt to reduce familywise error.” and “The sample size recruited was determined by the minimum of 190 required for conducting CFA on unidimensional models with factor loadings of .50 (66).”

3. Finally, in Table 5, it would be interesting if the authors could provide a test for comparing means to investigate the presence of statistically significant differences in means according to age.

Response: Unless we have misunderstood the reviewer’s meaning here (and our sincere apologies if we have), these analyses are already presented under the Results subsection “Age-related differences in trust beliefs”. We chose linear trend analysis because of a theoretical interest in the general trend rather than pairwise differences between particular age cohorts, and as this procedure has greater statistical power than between-groups ANOVA (Buckless & Ravenscroft, 1990. This higher power seems particularly important in this case for dealing with a smaller than ideal sample size to detect pairwise differences after correcting for error. Given the length of the manuscript, we have not described the pairwise differences in addition to the general trend, however, we would be happy to do so if required. These analyses reflect a similar pattern as the trend analyses; somewhat unsurprisingly considering the small effect sizes of the linear contrast analyses, means scores only differ by age on the General Peer Trust scale (using a between-groups ANOVA), of which Bonferroni-corrected pairwise comparisons reveal significant differences between 8- and 12-year-olds, and 9- and 12-year-olds.

4. The discussion is adequate, although it does not fully justify the need to provide a general measure of confidence based on the results obtained

Response: We have made some amendments to the General Discussion to clarify that the measures presented in this paper are unique from previous behavioural and self-report approaches in the constructs they aim to measure. For example on pg. 59, “However, the present findings demonstrate that unlike behavioural trust, general trust decreases during childhood and is likely a separable construct from the behaviour measured by the trust game that may measure more prosocial or economic behaviours”, and lines 1011-1013.

5. Reviewer #2: The article presents the development of a scale with a fairly solid design. It uses various samples, which, although they may be subject to criticism, I believe are adequate for the purpose of the research. I think that due to the complexity of the design, the article is difficult to follow, especially the methods and results.

Below, I outline some general guidelines to give coherence to the manuscript, and especially to make it easier to read for those who may not be familiar with scale validations. Personally, I am familiar with them, and I found it difficult to follow the rationale.

In the introduction, not all the analyses/hypotheses that will be carried out are presented. In fact, in Study Three, many things are introduced that were not mentioned beforehand. I believe that the CFA is precisely a strong point of the study, and the authors do not mention it.

Response: We have amended the General Introduction to flag earlier that internal structure and convergent validity will be examined, and that curvilinear regression analyses will also be conducted. We have also amended the various Analytical Procedures in line with the next comment. In particular, we have added the following text to the end of the General Introduction (on pg. 9) “, including whether any of these relationships are curvilinear. This final study also examined the internal structure through a confirmatory factor analysis (CFA), and further explored the evidence for convergent validity of the modified GTS and its alternate scales.”

6. In each methods section, the data analyses carried out should be presented. For example, in the more qualitative analysis, various indices or coefficients are mentioned without explaining what they are or how they were calculated. The same happens with the quantitative studies. It is not until the results section that it becomes clear what is going to be done, which makes the text difficult to follow.

After reviewing these general aspects, I will be able to provide more specific feedback on the issues in the manuscript. Below, I mention some specific areas for improvement.

Response: We apologise for the confusion in the wording of the analytic approach to Studies 1A and 1B. The numbers presented are not coefficients, but scores representing the proportions of responses coded as sufficiently meeting the definition of trust. We have amended the manuscript to clarify the description of these scores (see pg. 12). We have also amended the other Analytical Procedures, in particular, by moving the explanation of the CFA coefficients in Study 3 from the Results to the Methods (pg. 39).

7. (1) These studies also demonstrated the psychometric properties. ->“These studies also provides evidence of the psychometric quality of the scales. The psychometric properties can not be fully demonstrated (ever). In the same line, I suggest authors do not say “validate” but “provide evidence of validity” across the manuscript.

Response: Thank you for this feedback. We have amended the wording in line with this suggestion throughout the manuscript.

8. (2) Construct validity is a concept of the Standards for Psychological and Educational testing (1985) that was updated in 1999. Currently, we have “content validity”, “internal structure” and “relationship with external variables”. For concision (page 8), if I am not wrong, in the first study you provide test content and response process, in study 2, relationship with external variables (convergent/discriminant). In third study, you provide further evidence of relationship with external variables, in this case based on differences between relevant groups. I wonder if it is possible to provide evidence bases on the internal structure (i.e., factor analysis).

Response: We have amended the term “construct validity” throughout to clarify that we have conducted an examination of either the internal structure or convergent validity, as appropriate, and revised the manuscript to clarify earlier that CFA will be conducted to assess the internal structure of the scales.

9. (3) I think it is necessary to state where this study has been carried (I assume in Australia for the ethics office, but not sure). Did children or families receive any gratification for participation?

Response: Thank you for highlighting this omission. We now explicitly state that the samples in Studies 1A and 1B were Australian (see pp.11 and 16). Details concerning gratification for participation have been added to those relevant studies as well (1A and 3; pp. 11 and 35). For example, in Study 3 we have added: “and were offered to enter a prize drawer for one of ten AUD$20 vouchers.” (see pg. 35).

10. (4) I do not understand line 246. I think some verb or noun is missing.

Response: We have reworded the Analytical Procedure section of Study 1A to clarify this process, which now reads “A single author coded the definition responses dichotomously to indicate whether responses sufficiently described general trust, according to the definition provided above”.

11. (5) Design section I think is “Data analysis section”. More description is needed in order to make the study reproducible.

Response: Thank you for this suggestion. These sections have been renamed throughout the manuscript. Additional description has been added on pp. 12, 17, and 39.

12. (6) I am struggeling understanding Table 1/2. Some extra information about what this table presents (or adding information into a data analysis section). How is reliability measured?

Response: A codebook outlining the types of responses that were scored as describing each of these components has been added (as the updated S1 Appendix) to provide additional context for these tables. We have also clarified the wording in the Analytic Procedure for Study 1A to clarify further what this table represents (see pg. 12).

13. (7) In Study 1B is needed further discussion. As you are using undergraduate students they may have more knowledge that “general adults”.

Response: We have amended this discussion to include “While undergraduates typically have better linguistic understanding than the general population (Wild et al., 2022), limiting the generalisability of this finding, the similarity between educated adults and children further suggests evidence for the validity of using a downward-extension of the adult GTS measure.”.

14. (8) Authors are generating sum scores without providing evidence of internal structure. This is a problem since we do not have evidence that this can be done. As authors are using omega that depends on CFA, please report the results of CFA.

Response: McDonald’s omega internal reliability coefficients were generated using the standard scale reliability analysis tools available in jamovi, rather than as an effect size of the CFA. These analyses are reported separately in Study 3; we have amended the manuscript to clarify which software was used for each analysis (see pp. 13, 23, 39). Refer also to response #16, indicating the evidence for internal structure that demonstrates the appropriateness of generating sum scores.

15. (9) Table 4. Add means and SD

Response: Thank you for this suggestion. We had aimed to avoid repetition of results already presented in Table 3 (now labelled Table 6) but agree the information provides useful context. These descriptives have been added to what is now Table 7 (pg. 28).

16. (10) Here we finally find the CFA. This has not mentioned before, and I think it is necessary to present it. You need to further developed the estimators and correlation matrix used.

Response: We have revised the manuscript to flag earlier that CFA will be conducted, and in which study (e.g., pp 9, 32). However, concerning the development of estimators and correlation matrix, these elements were not needed for the model specification as we conducted a maximum likelihood CFA rather than analysing the unidimensional model using SEM. We have reported on the results of the CFA in the style of other recent scale development papers (e.g., Teunisse et al., 2020), and consider that further description of the factor structure would add little clarity when there is a strong theoretical basis for the scale’s structure. However, we would be happy to include this additional detail if required and leave it to the editor to decide.

17. (11) “Structural validity” does not exist “internal structure validity evidence”

Response: We have revised the use of this term throughout the manuscript to adopt “internal structure validity” and amended language that suggests “demonstrates validity” in favour of “provides evidence for the validity”, in line with your earlier suggestion.

18. (12) To assess for age differences, it is necessary to perform a multi-group confirmatory factor analysis before.

Response: A priori power analysis indicated samples of at least 100 children per age group would have been required to have confidence in the results of a multi-group CFA, and after several years of recruitment we were aware that sample size would not be possible to achieve. As the original adult version of the scales has had examinations of internal structure prior and its invariance confirmed across several cultures (e.g., Jasielska et al., 2021), examining its internal structure was not the primary focus of this paper. In cases such as this with such strong effect sizes, Kelly and Lai (2018) also recommend recruiting larger samples still, so we do not believe the data could attain sufficient power to conduct this analysis. We have amended the discussion of Study 3 to include this concern as a limitation: “Additionally, confidence in this finding is limited by our inability to assess measurement invariance between the age groups. Future research would benefit from recruiting a large enough sample to conduct multigroup confirmatory factor analysis, and examining whether age-related decreases in general and peer trust beliefs are more pronounced during earlier phases of development.” (see pg. 53).

19. (13) Which software was used?

Response: Thank you for highlighting that this information was not made available until the Results section of study 3. We have provided additional information about the software within the Analytical Procedure sections.

20. (14) IC of RMSEA is nearly impossible to be 95%: it is probably 90%

Response: Thank you for informing us of this error that we had overlooked. The results have been amended to reflect 90% confidence intervals.

21. (15) I suggest the authors to present chi/df and CFI, TLI, RMSEA in the same parenthesis.

Response: We have presented all CFA results within a single parenthetical for each. Thank you for your thoughtful feedback.

---

## [Editor Report · Decision Letter 1]

28 Mar 2025

Age-related differences in trust beliefs during middle childhood: Downward-extension and validation of the General Trust Scale

PONE-D-24-23774R1

Dear Dr. Wheeler,

We’re pleased to inform you that your manuscript has been judged scientifically suitable for publication and will be formally accepted for publication once it meets all outstanding technical requirements.

Kind regards,

Frantisek Sudzina

Academic Editor

PLOS ONE
---

## [Editor Report · Acceptance letter]

PONE-D-24-23774R1

PLOS ONE

Dear Dr. Wheeler,

I'm pleased to inform you that your manuscript has been deemed suitable for publication in PLOS ONE. Congratulations! Your manuscript is now being handed over to our production team.

Kind regards,

on behalf of

Dr. Frantisek Sudzina

Academic Editor

PLOS ONE